# Hepatitis C distribution across diverse population groups in the Eastern Mediterranean Region: An umbrella review

Zahra Abdolahinia[1,2], Sana Eybpoosh[3�*], Parya Jangipour Afshar[4], Ali Karamoozian[5], Kayhan Azadmanesh[6], Hamid Sharifi[1,7�*]

**1** HIV/STI Surveillance Research Center, and WHO Collaborating Center for HIV Surveillance, Institute for Futures Studies in Health, Kerman University of Medical Sciences, Kerman, Iran, **2** Department of Biostatistics and Epidemiology, Faculty of Public Health, Kerman University of Medical Sciences, Kerman, Iran, **3** Department of Epidemiology and Biostatistics, Research Centre for Emerging and Reemerging Infectious Diseases, Pasteur Institute of Iran, Tehran, Iran, **4** Health Services Management Research Center, Institute for Futures Studies in Health, Kerman University of Medical Sciences, Kerman, Iran, **5** Modeling in Health Research Center, Institute for Futures Studies in Health, Kerman University of Medical Sciences, Kerman, Iran, **6** Department of Molecular Virology, Pasteur Institute of Iran, Tehran, Iran, **7** Institute for Global Health Sciences, University of California, San Francisco, San Francisco, California, United States of America

উ These authors contributed equally to this work.
* sharifihami@gmail.com (HS); sana.eybpoosh@gmail.com (SE)

## Abstract

### Background

Hepatitis C virus (HCV) remains a major public health challenge in the Eastern Mediterranean Region (EMR), with prevalence varying across population groups. This umbrella review summarizes the distribution of HCV infection across diverse populations in the region.

### Methods

We conducted an umbrella review of systematic reviews with meta-analyses reporting HCV prevalence in the EMR. PubMed, Web of Science, Scopus, Iranian databases, including Magiran and the Scientific Information Database (SID), were searched. Google Scholar was additionally screened to ensure comprehensive coverage. Pooled estimates were extracted across four population groups: apparently healthy individuals; those with clinical or healthcare-associated exposure risk; patients co-infected with hepatitis viruses or other liver-related diseases; and key populations at increased risk (groups disproportionately affected due to their behaviors and higher vulnerability). For time-period analyses, meta-analyses were grouped into two intervals (before 2015 vs. 2015 and after) based on the median year of data collection of primary studies. Pooled prevalence was calculated using a random-effects model in STATA version 17.

**Data availability statement:** All relevant data are within the paper and its Supporting information files. The data were extracted from previously published studies, which are fully cited in the References. No new data were generated or collected in the course of this study.

**Funding:** This study was supported by the Iran National Science Foundation (INSF) and Kerman University of Medical Sciences (KMU) through a grant awarded to HSH (INSF Grant No. 4042461; KMU Grant No. 403000463). Additional support came from KMU in the form of salaries for ZA, PJA, AK, and HSH; the Pasteur Institute of Iran provided salaries for SE and KA. The specific roles of this author are articulated in the 'author contributions' section. The funders had no role in study design, data collection and analysis, decision to publish, or preparation of the manuscript.

**Competing interests:** The authors have declared that no competing interests exist.

## Results

A total of 55 meta-analyses were included. The pooled HCV prevalence was 31.0% (95% CI: 27.0–38.0) among key populations, and also 31.0% (95% CI: 12.0–49.0) among patients co-infected with other hepatitis viruses or liver-related diseases. Those with clinical or healthcare-associated exposure risk showed a prevalence of 28.0% (95% CI: 23.0–32.0), whereas the apparently healthy population had the lowest prevalence at 2.0% (95% CI: 1.0–2.0). Subgroup analysis indicated a decline in prevalence among clinically exposed populations after 2015, from 29% to 10%, coinciding with the direct-acting antivirals (DAAs) and strengthened infection-control practices. The highest prevalence was observed among key populations in Libya, healthcare-exposed populations in Morocco, and apparently healthy individuals in Egypt.

## Conclusion

Variation in HCV prevalence across populations in the EMR highlights the need for population-specific strategies to support progress toward World Health Organization elimination targets.

## Introduction

Hepatitis C virus (HCV) remains a significant global health burden, affecting a considerable number of people worldwide [1]. According to the World Health Organization (WHO) estimates, by 2024, approximately 50 million people will be living with chronic HCV infection globally, with about 1 million new infections occurring annually [2]. HCV infection is prevalent in all WHO regions, with the highest burden of disease in the Eastern Mediterranean Region (EMR), and 12 million chronically infected people. This is followed by South-East Asia and Europe, each with approximately 9 million cases, and the Western Pacific Region (7 million) [2]. The distribution of HCV genotypes varies across regions worldwide [3]. Genotypes 1, 2, and 3 are globally distributed, whereas genotypes 4 and 5 are found mainly in North Africa, and genotype 6 is predominantly distributed in Southeast Asia and southern China [3,4].

The epidemiological patterns of HCV in the EMR are highly diverse, with substantial variation in prevalence, risk factors, and circulating genotypes across countries [5,6]. As a blood-borne pathogen, HCV is primarily transmitted through exposure to infected blood, often due to unsafe medical injections, inadequate infection-control practices in healthcare settings, transfusion of unscreened blood, injection drug use, or sexual activities involving blood exposure [2]. The exceptionally high prevalence of HCV in Egypt has been largely attributed to historical mass treatment campaigns for schistosomiasis during the mid-20th century, in which reuse of injection equipment led to widespread iatrogenic transmission [7]. Although HCV infection has historically been a major indication for liver transplantation and reinfection after transplantation

was common, the epidemiological outlook has changed markedly following the introduction of highly effective direct-acting antivirals (DAAs) since 2013. These treatments have substantially reduced disease burden and mortality, lowering, though not eliminating the need for an effective vaccine [8]. Furthermore, HCV prevalence significantly differs across various populations, with higher rates observed among several populations such as people who inject drugs, hemodialysis patients, and people in prison [9–11].

Despite ongoing global and regional efforts, several contextual challenges continue to prevent HCV elimination in the EMR [12]. Although progress has been achieved in some countries, including improvements in governance and policy-making and expansion of HCV testing and treatment services, important gaps remain. Limited financial resources make it difficult to expand necessary interventions, including safe injection practices, harm reduction strategies, and broader access to diagnostic and therapeutic services. To accelerate progress toward elimination, these components need to be integrated into national plans for universal health coverage [12]. In line with this goal, the WHO introduced the global health strategy on viral hepatitis, defining elimination as a 90% reduction in incidence and a 65% reduction in hepatitis B and C–related mortality between 2015 and 2030 [13].

Several studies have examined HCV prevalence across EMR countries among diverse populations, including the general population, key populations, and individuals with clinical or healthcare-associated exposure risk [14,15]. However, available evidence remains fragmented, varies in scope and quality, and lacks systematic integration. To address this gap and identify both pooled prevalence estimates and the populations most affected, thereby informing future research and policies aimed at HCV elimination, this umbrella review provides a comprehensive overview of HCV prevalence across population groups in the EMR.

## Materials and methods

### Search strategy and study selection

The CoCoPop framework was used to identify keywords and guide the evaluation of articles retrieved from databases (S1 Table), incorporating the investigated Condition, Context, and Population relevant to this umbrella review of prevalence meta-analyses [16]. This umbrella review was reported in accordance with the PRISMA (Preferred Reporting Items for Systematic Reviews and Meta-Analyses) guidelines to ensure methodological transparency and completeness (Fig 1). In this paper, the term "reviews" refers to published systematic reviews that included meta-analyses and were eligible for inclusion. We also manually verified that no relevant primary studies had been omitted from the included meta-analyses. The term "primary studies" refers to original investigations reporting HCV prevalence in defined populations.

This study was conducted as part of a large project and was approved by the Ethics Committee of Kerman University of Medical Sciences, Kerman, Iran (Ethics Code: IR.KMU.REC.1403.319). We searched PubMed, Web of Science, Scopus, and Iranian databases, including Magiran and the Scientific Information Database (SID). Google Scholar was additionally screened to ensure comprehensive coverage. Searches included systematic reviews with meta-analyses reporting HCV prevalence in EMR countries published up to October 12, 2025, without language restrictions.

Meta-analyses reporting prevalence estimates with corresponding confidence intervals and providing sufficient data for their calculation were included. The literature search employed combinations of terms related to HCV and EMR countries, including alternative forms of country names, to ensure comprehensive coverage. Two authors (ZA and PJA) independently conducted the literature search. Reference lists of included reviews were also manually screened to identify additional relevant articles not captured in the initial search. The detailed search strategy is provided in the Supporting Information (S2 Table). Extracted information included author, publication year, number of included studies, number of events, sample size, HCV prevalence with 95% confidence intervals, and study population (S3 Table).

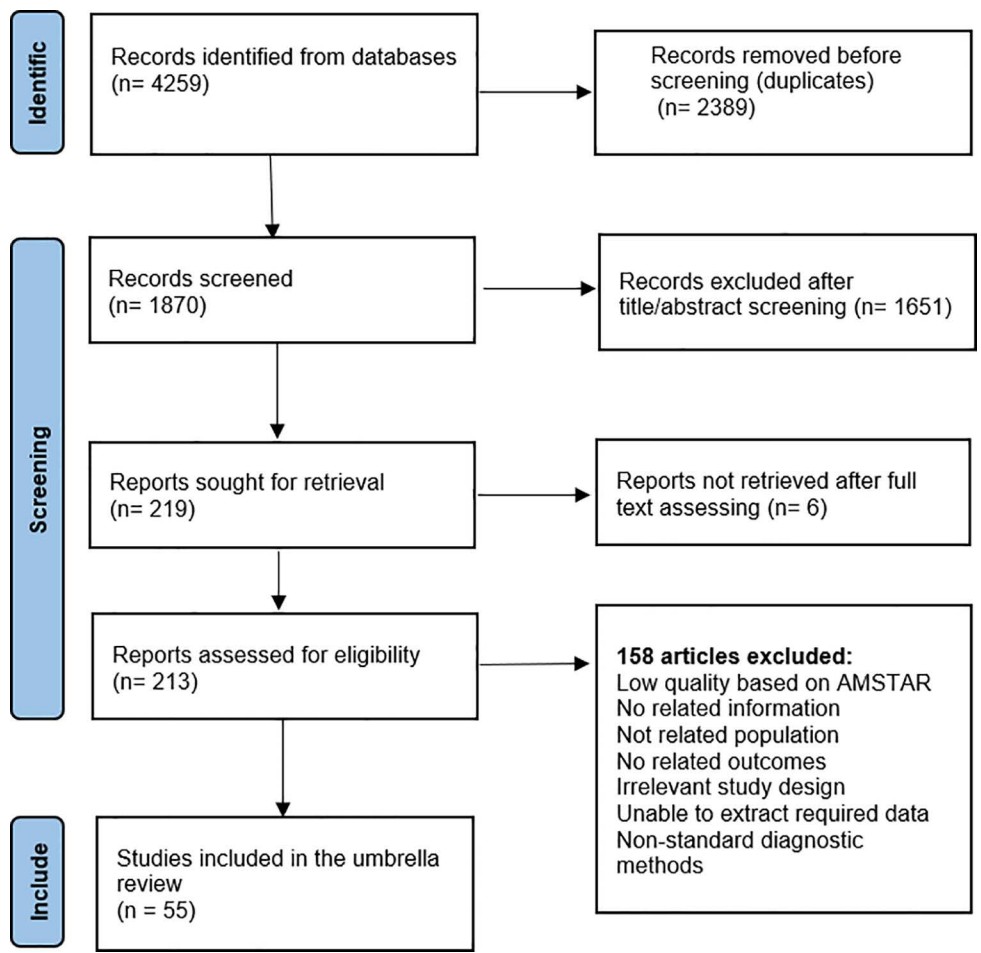

**Fig 1. Flowchart PRISMA of review selection for inclusion in the umbrella review.**

## Population categories and overlap assessment

Meta-analyses were stratified into four population groups to account for differences in exposure, risk, and clinical context. The first group comprised key populations with high behavioral and social risk, including people who inject drugs (PWID), people who use drugs, men who have sex with men (MSM), female sex workers (FSW; women who receive money or goods in exchange for sexual services), people living with HIV (PLHIV; individuals diagnosed with HIV), and street children (children and adolescents who live and/or work on the streets with limited or no parental care) [17]. The second group included individuals with clinical or healthcare-associated exposure risk, such as patients undergoing hemodialysis, those with thalassemia, hemophilia, or other inherited coagulation disorders, multi-transfused patients, healthcare workers, household contacts of HCV-infected individuals, and patients with other comorbidities. The third group comprised patients with liver-related conditions or hepatitis B (HBV) co-infection. The fourth group consisted of apparently healthy individuals representing the general community, including participants in screening programs or household surveys, blood donors, pregnant women, children, refugees, and military recruits.

To minimize overlap among primary studies and facilitate more accurate country-level comparisons within the EMR, we included meta-analyses reporting pooled estimates at the country level. Overlap of primary studies across systematic reviews was assessed using citation matrices, where rows represented unique primary studies and columns represented

reviews. Cells indicated the presence [1] or absence (0) of each study in a given review. For each population category (key populations, co-infected patients, individuals with clinical or healthcare-associated exposure risk, and apparently healthy individuals), the corrected covered area (CCA) was calculated using the following formula:

$$CCA = \frac{N_r - N_s}{N_s(N_k - 1)} \times 100$$

Where $N_r$ as the total number of study occurrences across reviews, Ns is the number of unique studies, and $N_k$ as the number of reviews. Based on published recommendations, we interpreted CCA values ≤5% as slight, 6–10% as moderate, 11–15% as high, and >15% as very high overlap [18].

Screening of all publications based on eligibility criteria was performed independently by two authors (ZA and PJA). In the first step, all titles and abstracts imported into EndNote were screened. In cases of disagreement between the two authors, a third reviewer was consulted to reach a final decision. A standardized data extraction form was created in an Excel spreadsheet to efficiently collect information from the included studies. The following data were extracted: first author, year of publication, population category, number of events and participants, number of studies included in the meta-analysis, prevalence of HCV, and 95% CI (S3 Table).

### Quality assessment

To assess the methodological quality of the included systematic reviews, two authors (ZA and PJA) independently evaluated the articles using the original AMSTAR tool (A Measurement Tool to Assess Systematic Reviews), which comprises 11 items assessing key aspects of review methodology, as described by Shea et al. [19]. Each item was rated as 'Yes', 'No', or 'Not applicable', yielding a total score ranging from 0 to 11. Scores of 8–11 were classified as high quality, 4–7 as moderate quality, and 0–3 as low quality. Any discrepancies between the reviewers were resolved through discussion. Studies with sufficient quality were selected for further investigation. The quality ratings for each study are provided in Supplementary information (S3 Table). Only studies with moderate or high quality (AMSTAR score ≥4) were included in subsequent analyses, whereas low-quality studies (AMSTAR score 0–3) were excluded. Excluding low-quality meta-analyses reduces potential bias and ensures that the findings of this umbrella review are based on more reliable evidence.

### Data synthesis

Random-effects models were used to estimate the pooled prevalence and corresponding 95% confidence intervals (CIs). Heterogeneity across studies was evaluated using the I² statistic and Cochran's Q test, with substantial heterogeneity defined as I² > 50% or a Q test p-value <0.10, in line with established recommendations for meta-analyses of observational studies (Higgins & Thompson, 2002; Huedo-Medina et al., 2006) [20]. P-values <0.05 were considered statistically significant. Detailed heterogeneity and pooled prevalence estimates are provided in Supplementary information (S3 Table). Publication bias was not assessed using formal tests or funnel plots because these are unreliable for meta-analyses of proportions due to the bounded nature and instability of variance in proportion data. Instead, publication bias was evaluated qualitatively, consistent with methodological recommendations for prevalence meta-analyses [21]. The random-effects model accounts for expected heterogeneity across countries with varying population sizes and prevents studies with very large sample sizes from disproportionately influencing the pooled estimate. All analyses were conducted using Stata version 17.0 (StataCorp, College Station, TX, USA).

### Time-period subgroup analysis

To explore temporal trends in HCV prevalence in the EMR, we conducted a subgroup analysis based on the median year of data collection of the primary studies included in each meta-analysis. Because umbrella reviews synthesize evidence from existing meta-analyses, the publication year of a meta-analysis does not necessarily reflect the time frame of the

underlying data. Using the median year of data collection provides a robust measure of the central temporal distribution while minimizing the influence of early or late outlier studies. For each meta-analysis, we extracted the year of data collection for all primary studies (or, if unavailable, the year of publication or last reported search date). The median year of the included primary studies was then calculated and used as the temporal indicator. Meta-analyses were classified into two time periods: before 2015 vs. 2015 and after, reflecting the expansion of direct-acting antiviral (DAA) treatments. Subgroup analyses compared pooled prevalence estimates across these periods to assess temporal changes in HCV epidemiology within the region.

## Results

A total of 4,259 records were identified through database searches. After removing 2,389 duplicates, 1,870 records remained for screening. Following the title and abstract assessment, 1,651 records were excluded, leaving 219 records for full-text retrieval. Six records could not be retrieved, and the full texts of the remaining 213 records were assessed for eligibility. Of these, 158 were excluded due to irrelevant content, resulting in 55 meta-analyses included in the study. When a study reported separate prevalence estimates for different countries or population categories, each estimate was treated as an individual observation. Consequently, although 55 studies were included, a total of 218 prevalence estimates were extracted and analyzed across EMR countries. These estimates were distributed across the following population groups: key populations (including PWID, people who use drugs, MSM, FSW, PLHIV, and street children) contributed 76 estimates (35.9%; references [22–48]); co-infected patients contributed 9 estimates (4.2%; references [43,49–52]); individuals with clinical or healthcare-associated exposure contributed 78 estimates (36.8%; references [23,26,43,44,49,51,53–73]; and apparently healthy individuals contributed 49 estimates (23.1%; references [6,25,26,28,40,43,44,49,54–57,64,66,68,72,74–85]). The Frequency of prevalence estimates by EMR countries is shown in Fig 2. Iran, Pakistan, and Egypt accounted for the majority of available estimates, respectively, which indicates a higher research focus or disease burden in these countries compared with others in the region.

The I² values reported across included meta-analyses ranged from 23.4% to 100%, indicating varying levels of heterogeneity among the pooled estimates.

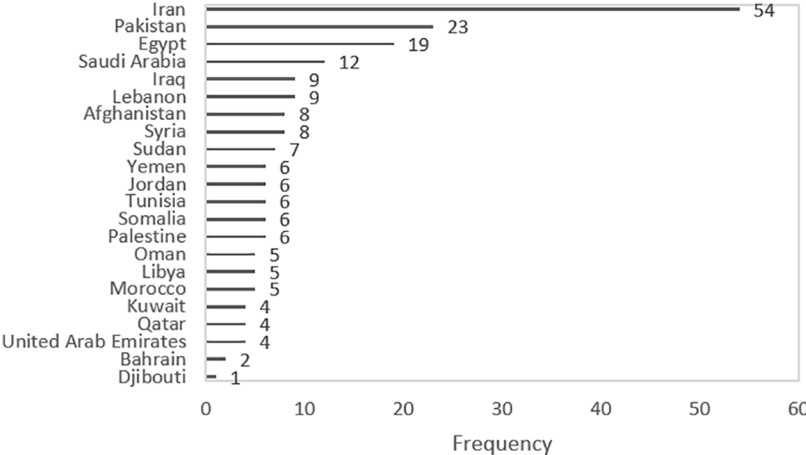

**Fig 2. Frequency of HCV prevalence estimates by countries in the Mediterranean region up to 2025.**

## Overlap assessment

The extent of overlap between systematic reviews varied across population categories. The lowest overlap was observed among apparently healthy individuals, including the general population, screening samples, blood donors, pregnant women, children, refugees, household surveys, and army recruits, with a CCA of 4.8%. In key populations, the CCA reached 5.4%, indicating moderate overlap. Individuals with clinical or healthcare-associated exposure risk had a CCA of 7.4%, indicating moderate overlap. The highest overlap was observed among co-infected patients, with a CCA of 9.0%.

## HCV prevalence in EMR

A total of 19 studies (49 estimates), including 65,900,000 participants, assessed pooled HCV prevalence among apparently healthy individuals. The pooled prevalence in this group was 2.0% (95% CI: 1.0–2.0) (Fig 3). The pooled prevalence of HCV in 21 related studies (78 estimates) from people with clinical or healthcare-associated exposure risk, comprising 625,828 participants, was 28.0% (95% CI: 23.0–32.0) (Fig 4). In the co-infected patient group, five related studies (providing 9 estimates) included 155,980 participants, and the prevalence of HCV was 31.0% (95% CI: 12.0–49.0) (Fig 5). Finally, 25 studies (76 estimates) among key population groups, 22,200,000 participants were assessed, and the HCV prevalence in this group was 31.0% (95% CI: 27.0–38.0) (Fig 6). Prevalence estimates were tabulated by population category and country. Forest plots were used to visually display the pooled prevalence and 95% CI for each group (Figs 3–6).

HCV prevalence across EMRO countries varied markedly by population group. In apparently healthy individuals (Panel A), Egypt, Pakistan, and Yemen had the highest prevalence. Among populations with clinical or healthcare-associated exposures (Panel B), prevalence was highest in Morocco, Syria, and Saudi Arabia. In key populations (Panel C), Libya, Morocco, and Saudi Arabia ranked highest, respectively. These findings highlight substantial heterogeneity in HCV prevalence across EMR countries, driven by population-specific risk profiles (Fig 7).

## Time period

A temporal shift in HCV prevalence was observed among patients with clinical or healthcare-associated exposure risks, where HCV prevalence declined from 29% (95% CI: 24%–34%) before 2015 to 10% (95% CI: 7%–12%) in 2015 and thereafter. In contrast, no meaningful reduction was observed among key populations or apparently healthy individuals (S1–S6 Figs). Insufficient data were available for co-infected groups to assess the temporal shift.

## Discussion

This review provided evidence on the distribution of HCV across different population groups in the EMR. In the key population and also other hepatitis types and liver-related co-infected patients, the pooled prevalence was relatively higher than in other groups (31.0%). This estimate was lower among people with clinical or healthcare exposure risk (28.0%), and particularly among apparently healthy individuals (2.0%). Also, HCV prevalence varies among EMR countries. Egypt, Pakistan, and Yemen exhibited the highest prevalence of HCV in apparently healthy individuals. Among populations with clinical or healthcare-related exposures, prevalence was highest in Morocco, Syria, and Saudi Arabia. Within key populations, Libya, Morocco, and Saudi Arabia had the highest prevalence.

According to the WHO reports, settings with intermediate to high HCV prevalence are defined as those with an HCV antibody seroprevalence exceeding 2% in the general population [86]. Based on our findings, some EMR countries fall into this category, thereby classifying the region as having one of the highest prevalence rates worldwide. This categorization highlights the persistent burden of infection in the EMR compared to countries in the European Union/European Economic Area (EU/EEA), where the prevalence estimates among the apparently healthy individuals and general population are markedly lower [87]. It should be noted that European countries have taken blood screening and iatrogenic infection control (e.g., during dialysis or injections) seriously since the 1990s [88,89]. Such cases are likely to contribute to explaining the differences between regions.

| Study | | Effect size with 95% CI | Weight (%) |
|---|---|---|---|
| Obeid et al, 2024 | | 0.02 [ 0.02, 0.02] | 2.19 |
| Obeid et al, 2024 | | 0.00 [ -0.00, 0.00] | 2.18 |
| Abbasi et al , 2023 | | 0.07 [ 0.05, 0.09] | 1.72 |
| Abbasi et al , 2023 | | 0.03 [ 0.03, 0.04] | 2.15 |
| Abbasi et al , 2023 | | 0.09 [ 0.07, 0.10] | 1.86 |
| Abbasi et al , 2023 | | 0.01 [ -0.00, 0.02] | 2.11 |
| Abbasi et al , 2023 | | 0.00 [ 0.00, 0.00] | 2.19 |
| Muhammad et al, 2022 | | 0.03 [ 0.02, 0.03] | 2.15 |
| Al Kanaani et al, 2018 | | 0.05 [ -0.08, 0.19] | 0.25 |
| Ghaderi et al, 2017 | | 0.06 [ 0.03, 0.08] | 1.67 |
| Ghaderi et al, 2017 | | 0.02 [ 0.01, 0.02] | 2.15 |
| Ghaderi et al, 2017 | | 0.00 [ 0.00, 0.00] | 2.19 |
| Ghaderi et al, 2017 | | 0.00 [ -0.00, 0.01] | 2.16 |
| Ghaderi et al, 2017 | | 0.00 [ -0.01, 0.02] | 2.06 |
| Ghaderi et al, 2017 | | 0.01 [ 0.01, 0.01] | 2.19 |
| Ghaderi et al, 2017 | | 0.01 [ 0.00, 0.01] | 2.17 |
| Ghaderi et al, 2017 | | 0.01 [ -0.00, 0.01] | 2.14 |
| Kasraian et al, 2025 | | 0.00 [ 0.00, 0.00] | 2.19 |
| Sarwat et al, 2019 | | 0.06 [ 0.05, 0.07] | 2.15 |
| Kouyoumjian et al, 2017 | | 0.12 [ 0.11, 0.13] | 2.13 |
| Al Kanaani et al, 2017 | | 0.00 [ 0.00, 0.00] | 2.19 |
| Mirminachi et al, 2017 | | 0.01 [ 0.00, 0.01] | 2.18 |
| Hassan-Kadle et al, 2018 | | 0.05 [ 0.03, 0.07] | 1.79 |
| Bagheri Amiri et al, 2016 | | 0.31 [ 0.18, 0.44] | 0.25 |
| Chaabna et al, 2016 | | 0.01 [ 0.00, 0.02] | 2.12 |
| Chaabna et al, 2016 | | 0.01 [ 0.00, 0.02] | 2.11 |
| Chaabna et al, 2016 | | 0.02 [ 0.01, 0.02] | 2.15 |
| Chaabna et al, 2016 | | 0.00 [ 0.00, 0.01] | 2.18 |
| Chemaitelly et al, 2015 | | 0.00 [ 0.00, 0.00] | 2.19 |
| Chemaitelly et al, 2015 | | 0.00 [ 0.00, 0.01] | 2.18 |
| Chemaitelly et al, 2015 | | 0.00 [ 0.00, 0.00] | 2.19 |
| Chemaitelly et al, 2015 | | 0.00 [ 0.00, 0.00] | 2.19 |
| Chemaitelly et al, 2015 | | 0.00 [ 0.00, 0.00] | 2.19 |
| Chemaitelly et al, 2015 | | 0.01 [ 0.00, 0.01] | 2.18 |
| Fadlalla et al, 2015 | | 0.01 [ 0.00, 0.01] | 2.18 |
| Fadlalla et al, 2015 | | 0.01 [ 0.00, 0.01] | 2.19 |
| Mohamoud et al, 2016 | | 0.02 [ 0.01, 0.02] | 2.18 |
| Mohamoud et al, 2016 | | 0.00 [ -0.00, 0.01] | 2.14 |
| Mohamoud et al, 2016 | | 0.01 [ -0.00, 0.01] | 2.14 |
| Mohamoud et al, 2016 | | 0.00 [ -0.00, 0.01] | 2.17 |
| Mohamoud et al, 2016 | | 0.00 [ 0.00, 0.00] | 2.19 |
| Mohamoud et al, 2016 | | 0.02 [ 0.01, 0.02] | 2.18 |
| Mohamoud et al, 2016 | | 0.01 [ 0.01, 0.02] | 2.12 |
| Mohamoud et al, 2016 | | 0.00 [ 0.00, 0.00] | 2.19 |
| Mohamoud et al, 2016 | | 0.01 [ 0.00, 0.02] | 2.14 |
| Mohamoud et al, 2016 | | 0.02 [ 0.01, 0.02] | 2.13 |
| Ahmed et al, 2025 | | 0.05 [ 0.03, 0.07] | 1.87 |
| Sharifi Ali Mude et al, 2025 | | 0.04 [ 0.02, 0.05] | 1.88 |
| Khodabandeh et al, 2013 | | 0.01 [ 0.00, 0.01] | 2.19 |
| **Overall** | | 0.02 [ 0.01, 0.02] | |

Heterogeneity: $\tau^2 = 0.00$, $I^2 = 99.96\%$, $H^2 = 2505.28$

Test of $\theta_i = \theta_j$: $Q(48) = 6470.31$, $p = 0.00$

Test of $\theta = 0$: $z = 5.00$, $p = 0.00$

0.000    0.200    0.400

Random-effects REML model

**Fig 3. Forest plot of prevalence estimates for the apparently healthy individuals' group.**

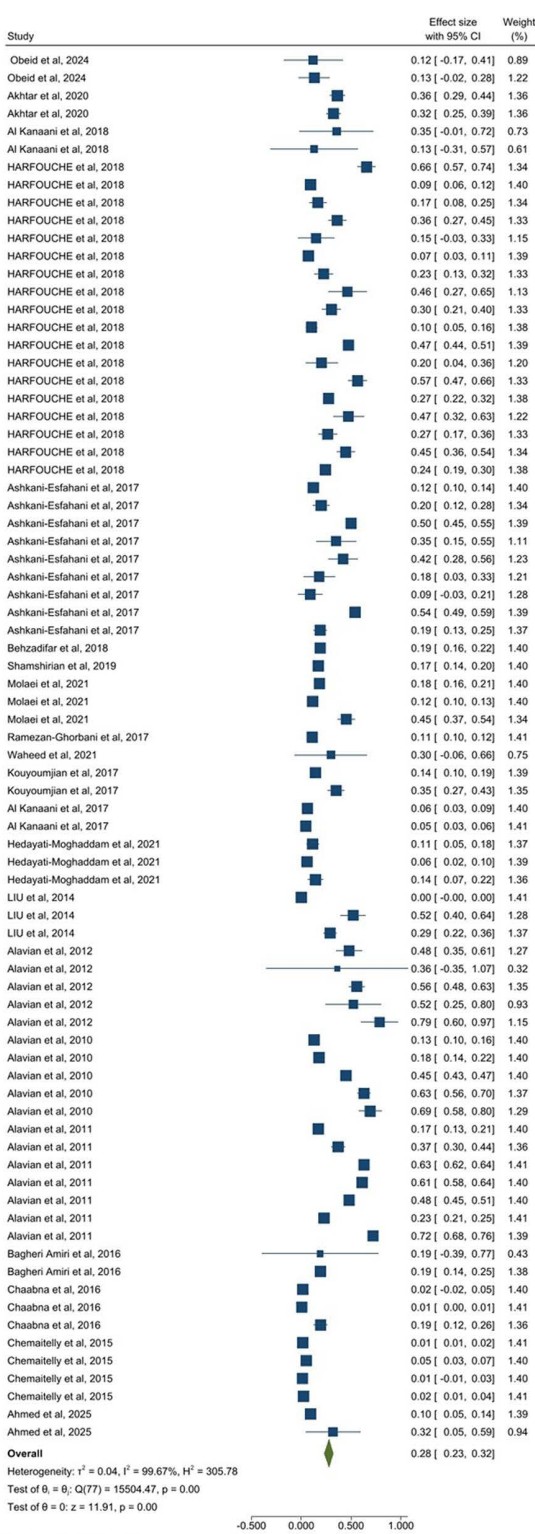

**Fig 4. Forest plot of prevalence estimates for people with clinical or healthcare-associated exposure risk.**

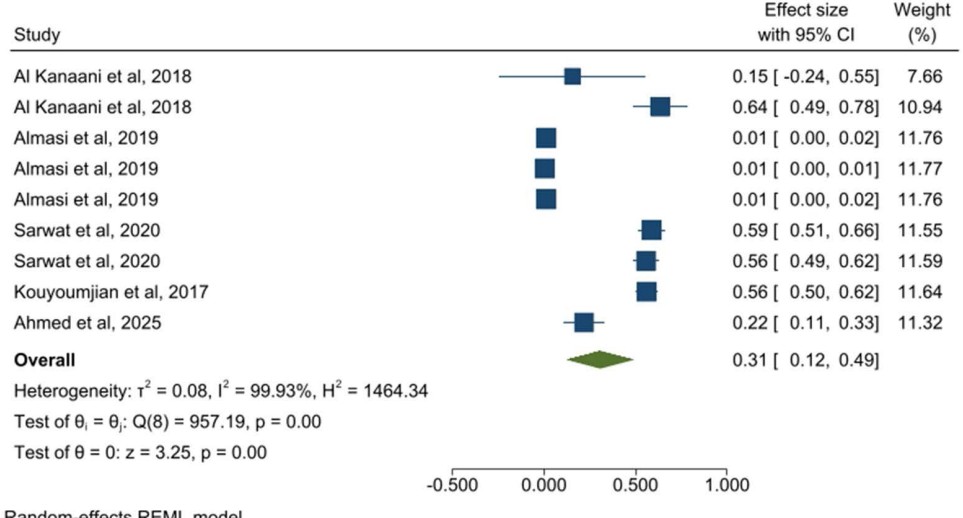

**Fig 5. Forest plot of prevalence estimates for the co-infected patients' group.**

The high prevalence of HCV in our study was observed among key populations. These groups are at elevated risk due to behavioral and social factors such as unsafe injection practices, sharing of contaminated equipment, co-infection with HIV, high-risk sexual behaviors, and inadequate infection control within prisons. Although there are few studies on the prevalence of HCV in Libya, in line with this study, a meta-analysis showed that the prevalence of HCV among PWID in this country is high, and HCV was most prevalent among PWID. Importantly, bordering areas with a high prevalence of the disease, such as Egypt, increase the country's vulnerability, and political challenges and ongoing healthcare issues could increase the risk of transmission [90]. Furthermore, limited access to preventive and harm-reduction services increases the transmission risk in these populations, highlighting the need for targeted public health interventions within the region [24,91].

Despite the availability of effective DAAs, PWID often face barriers, including stigma, discrimination, and inconsistent health services [92]. Implementing comprehensive harm-reduction programs such as needle and syringe programs and opioid substitution therapy (OST) can reduce transmission and improve treatment outcomes. Also, adapted interventions to the social and cultural context of each country will be necessary to reach this key population effectively. For instance, harm-reduction programs, access to DAAs for everyone, and integrating HIV/hepatitis services are also effective steps to reduce the burden of disease in this region, especially in the high-risk population [93].

From an epidemiological perspective, healthcare-related exposures play a major role in HCV transmission. In addition to blood transfusions, a range of unsafe medical practices, such as hemodialysis, surgical procedures, organ transplantation, and injections administered in both clinical and household settings, have contributed to the spread and overall burden of HCV infection. Both patients and healthcare workers are at risk; patients may be exposed through contact with contaminated medical equipment, while healthcare workers face occupational hazards, particularly accidental blood exposure. Although transmission from infected healthcare workers to patients is rare, such events are documented [94].

As highlighted in previous studies, countries with a very high population-level prevalence, such as Egypt, have been strongly affected by care-associated transmission, whereas countries with a lower rate in EMR or other regions showed minimal healthcare-related risk [95,96]. It should be noted that studies in Egypt have revealed a significant decrease in HCV prevalence over time in the general population, reflecting improved infection control practices, blood screening, and the implementation of national treatment campaigns [97,98].

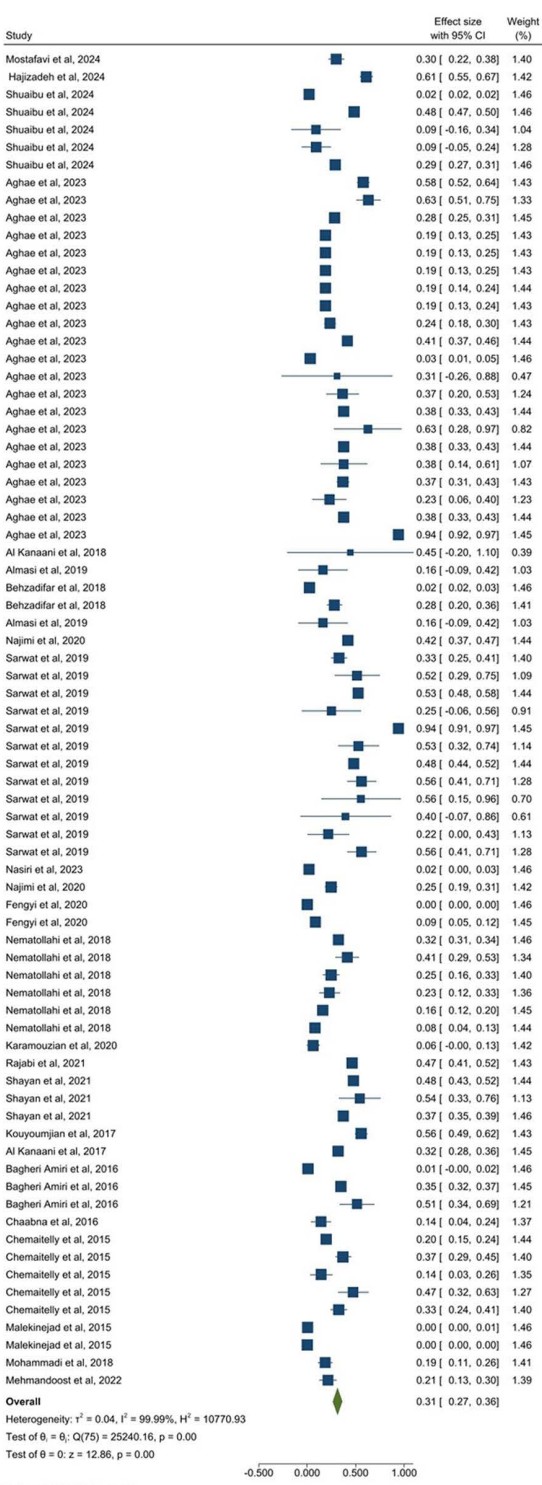

**Fig 6. Forest plot of the prevalence estimates for key population groups.**

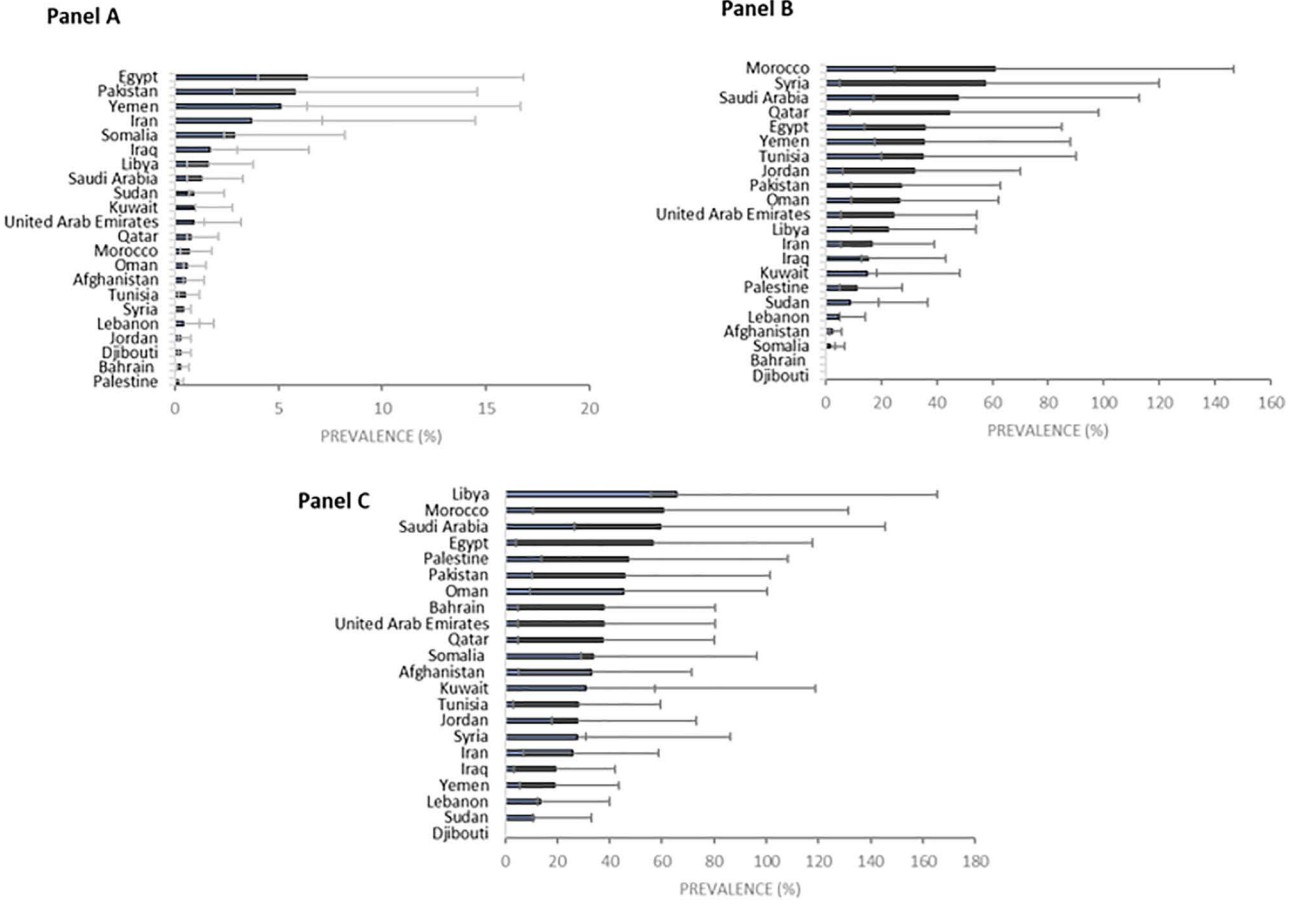

**Fig 7. Rank of EMR countries in HCV prevalence in apparently healthy individuals (Panel A), people with clinical or healthcare-associated exposure risk (Panel B), and the key population (Panel C).**

On the other hand, Pakistan faces a major public health challenge due to the widespread use of unnecessary injections, which has contributed to the high prevalence of HCV. As a result, populations that are generally considered apparently healthy, such as blood donors, have shown significant infection rates. In our study, published based on a meta-analysis, Pakistan ranked second in the region in terms of HCV prevalence among apparently healthy individuals, highlighting the impact of healthcare-associated exposures even in low-risk groups. Notably, a study in Morocco found that blood transfusions, dental care, and surgical history were strongly associated with HCV infection [99]. These evidences suggest that in settings with prevalent exposure to pathogens, community-level transmission can be significant even in groups without traditional behavioral risk factors, and emphasizes the need for infection control measures and targeted public health interventions in both clinical and general populations.

Our subgroup time period analysis showed that in 2015 and after, which coincided with a time when the number of people treated with DAAs increased significantly, and various countries began to plan and expand treatment, the HCV prevalence decreased in the clinical and health care population group. In line with our findings, several countries in the EMR have made significant progress in controlling HCV, with Egypt being the most prominent example. Large-scale national screening programs, widespread access to DAA, improved injection safety, and strengthened blood screening have significantly reduced the incidence, placing Egypt as a global leader in HCV elimination efforts [100]. However, despite these

gains, our findings still show that Egypt has the highest HCV prevalence in the region. The historical burden of infection explains this apparent contradiction. For decades, Egypt has experienced exceptionally high transmission rates, resulting in a large number of people who were infected long before recent interventions were implemented. Thus, although current transmission has declined sharply, the accumulated reservoir of chronic infections continues to maintain high national prevalence estimates. This highlights an important distinction between successfully reducing incidence and achieving a slower, long-term reduction in overall population-level prevalence [101,102].

Systemic weaknesses in EMR health systems, as noted in the WHO Vision 2023 report [103], explain the high HCV prevalence in populations. EMR countries face ongoing humanitarian crises, armed conflicts, and limited healthcare resources, all of which hinder the consistent implementation of infection-control practices. Overcrowded facilities, insufficiently trained personnel, and fragmented surveillance further increase the risk of nosocomial transmission. While some countries in the region have achieved notable public health successes, these gains remain variable, and many populations continue to be exposed to unsafe medical practices. The compounded effects of conflict and natural disasters have strained health services, emphasizing the critical need for strengthened infection control, capacity building, and effective oversight systems to reduce iatrogenic HCV transmission in the region [103]. Although DAAs have greatly improved outcomes, the region still faces barriers to care. Limited access to DAAs, weak health system infrastructures, and high prevalence of high-risk groups such as PWID contribute to a considerable burden of HCV, which should be considered [104,105].

## Limitation

This umbrella review has several limitations. Although our findings indicate variation in HCV prevalence across EMRO countries, it is important to consider that the number and quality of available studies differ substantially between countries. Nations such as Egypt, Iran, and Pakistan have conducted numerous epidemiological investigations with large sample sizes, while others, including Somalia and Djibouti, have limited data due to political instability, inadequate surveillance systems, and restricted research capacity [106]. As a result, it may partially contribute to observed differences in infection burden between regions. Additionally, overall low-to-moderate CCA values suggest limited overlap among included reviews; some primary studies, especially in co-infected populations, appeared in multiple reviews, warranting cautious interpretation. Finally, meta-regression indicated no significant country-level effect; however, these results should be interpreted cautiously due to possible limited power to detect small differences between countries. Despite these issues, this umbrella review suggested a comprehensive overview of HCV distribution across EMR population groups.

## Conclusion

Despite the high burden of HCV in the EMR, data on prevalence across population groups remain sparse. This comprehensive review provides a synthesis of the available evidence and provides important insights into population-specific risks and patterns of virus circulation, which are essential for targeted public health interventions and policy-making in the region. While overall prevalence remains low among apparently healthy individuals, higher rates have been observed among key populations, such as PWID, people in prison, and FSW, as well as among individuals exposed to clinical or healthcare-associated factors. These disparities highlighted the influence of risk factors on HCV transmission dynamics in the region. Key steps seem to be increasing access to DAAs, improving infection-control practices in healthcare settings, and providing harm-reduction programs for PWID. Combining HCV screening and treatment with HIV and other services, along with strong surveillance systems, will help identify and manage co-infections more effectively. Policymakers would be better to focus resources on high-risk populations and work together regionally to share best practices and support countries facing health system challenges. These measures are necessary for making real progress toward the WHO's hepatitis elimination goals and highlight the need for targeted prevention strategies in different populations.

                                                                 

## Supporting information

**S1 Table. Condition/Context/Population criteria for inclusion of studies.**
(DOCX)

**S2 Table. Database search strategy using PubMed, Scopus, and Web of Science.**
(DOCX)

**S3 Table. Summary of reviews included in the Umbrella Review of HCV prevalence across diverse populations in EMR.**
(DOCX)

**S1 Fig. Forest plot of prevalence estimates for the apparently healthy individuals' group before 2015.**
(TIF)

**S2 Fig. Forest plot of prevalence estimates for the apparently healthy individuals' group in 2015 and after.**
(TIF)

**S3 Fig. Forest plot of prevalence estimates for people with clinical or healthcare-associated exposure risk before 2015.**
(TIF)

**S4 Fig. Forest plot of prevalence estimates for people with clinical or healthcare-associated exposure risk in 2015 and after.**
(TIF)

**S5 Fig. Forest plot of the prevalence estimates for key population groups before 2015.**
(TIF)

**S6 Fig. Forest plot of the prevalence estimates for key population groups in 2015 and after.**
(TIF)

## Acknowledgments

**AI declaration:** An AI-based tool was used to improve grammar and language clarity; **no** content generation or data analysis was performed using artificial intelligence.

## Author contributions

**Conceptualization:** Hamid Sharifi, Zahra Abdolahinia, Sana Eybpoosh, Ali Karamoozian, Kayhan Azadmanesh.

**Data curation:** Zahra Abdolahinia, Parya Jangipour Afshar.

**Formal analysis:** Zahra Abdolahinia.

**Investigation:** Hamid Sharifi, Zahra Abdolahinia, Sana Eybpoosh, Ali Karamoozian, Kayhan Azadmanesh.

**Methodology:** Hamid Sharifi, Zahra Abdolahinia, Sana Eybpoosh, Ali Karamoozian, Kayhan Azadmanesh.

**Project administration:** Hamid Sharifi, Sana Eybpoosh.

**Software:** Zahra Abdolahinia.

**Supervision:** Hamid Sharifi.

**Validation:** Hamid Sharifi, Sana Eybpoosh, Kayhan Azadmanesh.

**Visualization:** Zahra Abdolahinia.

**Writing – original draft:** Zahra Abdolahinia.

**Writing – review & editing:** Hamid Sharifi, Zahra Abdolahinia, Sana Eybpoosh, Parya Jangipour Afshar, Ali Karamoozian, Kayhan Azadmanesh.

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
