## [Decision Letter · Decision Letter 0]

10 Feb 2026

Dear Dr. Sharifi,

Thank you for submitting your manuscript to PLOS ONE. After careful consideration, we feel that it has merit but does not fully meet PLOS ONE’s publication criteria as it currently stands. Therefore, we invite you to submit a revised version of the manuscript that addresses the points raised during the review process.

We look forward to receiving your revised manuscript.

Kind regards,

Ashraf Elbahrawy

Academic Editor

PLOS One

Journal Requirements:

2. Please include a separate caption for each figure in your manuscript.

Reviewers' comments:

Reviewer's Responses to Questions

**Comments to the Author**

1. Is the manuscript technically sound, and do the data support the conclusions?

Reviewer #1: Yes

Reviewer #2: Yes

2. Has the statistical analysis been performed appropriately and rigorously?

Reviewer #1: Yes

Reviewer #2: Yes

3. Have the authors made all data underlying the findings in their manuscript fully available?

Reviewer #1: No

Reviewer #2: Yes

4. Is the manuscript presented in an intelligible fashion and written in standard English?

Reviewer #1: No

Reviewer #2: Yes

Reviewer #1: Abstract

• Clarify: this is a systematic review, not a general review.

⸻

Background

• Line 34: “Key populations at increased risk” should be briefly clarified.

⸻

Introduction

• Line 70: Clarify the meaning of “incarcerated individuals.”

• Line 72: EMRO – clarify what it stands for; check if “O” is added by mistake.

⸻

Methods

• Lines 117–118: Clarify definitions of female sex workers (FSW), people living with HIV, and street children, and add supporting references.

• Line 141: Add “of HCV” after prevalence.

• Line 145: Complete all 11 items (currently only 5 mentioned).

• Line 154: Clarify meaning of p < 0.10 (substantial heterogeneity defined as I² > 50% or p < 0.10 for the Q test).

⸻

Results

• Line 188: Clarify meaning of (key population(20–46)).

• Line 201: Correct interpretation: “CCA reached 5.4%, indicating moderate overlap” (not slight).

• Line 226: Co-infected group not mentioned in the temporal shift in prevalence — add if relevant.

⸻

References

• Numbers 13, 46, and 58: Add full citation details (volume, page numbers, etc.).

⸻

English editing

• Needed throughout. Example: Line 317 –

“As a result, may partially contribute to observed differences in infection burden” → needs grammatical/clarity editing.

⸻

Tables

• Tables are not included in the review — consider including them or clarifying.

Reviewer #2: The manuscript is methodologically sound, timely, and relevant to public health in a high-burden region.

Main issues requiring attention (major comments):

Inconsistency in time-period grouping definition

Abstract says “median year of meta-analysis publication”; methods say “median year of data collection of primary studies”. → Align and clarify which one was actually used.

Search date clarification

Stated search up to October 12, 2025 (likely a typo for 2024 or earlier). Confirm and correct.

Minor numerical discrepancies / rounding

e.g., key populations reported as 32% in one place, 31% in another → standardize presentation (preferably as percentages with consistent CI formatting).

Quality assessment details

Explicitly state AMSTAR threshold for inclusion (e.g., ≥4) and discuss implications of excluding low-quality meta-analyses.

Heterogeneity & publication bias

Inconsistent use of “EMR” vs “EMRO”.

.

Reviewer #1: **Yes:** Sadek Mostafa Sadek AbdelaleemSadek Mostafa Sadek AbdelaleemSadek Mostafa Sadek AbdelaleemSadek Mostafa Sadek Abdelaleem

Reviewer #2: **Yes:** Ali MadianAli MadianAli MadianAli Madian

---

## [Author Response · Author response to Decision Letter 1]

23 Feb 2026

PLOS One

February 16, 2026

Dear Dr. Emily Chenette

Ref: Submission ID: PONE-D-26-00865

Re: "Hepatitis C distribution across diverse population groups in the Eastern Mediterranean Region: An Umbrella Review."

Thank you for your interest in this submission and for providing us with constructive and helpful feedback. We accepted all the suggestions and applied the requested edits. We have done our best to address every point made by the reviewers. Below, we provide point-by-point responses to all editor and reviewer comments. All changes made in the revised manuscript are highlighted, and relevant supporting information has been updated accordingly.

We hope this response will be viewed favorably and the manuscript will be considered for publication in your worthy journal.

Best Regards

Hamid Sharifi

Professor in Epidemiology

HIV/STI Surveillance Research Center,

and WHO Collaborating Center for HIV Surveillance,

Kerman University of Medical Sciences,

Kerman, Iran

A. Academic Editor comments

1. Ensure the manuscript follows PLOS ONE style requirements and correct file naming conventions.

Response: Thanks for your consideration. The manuscript has been revised to comply with PLOS ONE formatting requirements, and all files have been named according to the journal guidelines.

2. Provide a separate caption for each figure.

Response: Separate captions have been provided for all figures in the manuscript.

3. Review any reviewer-suggested references and cite them only if relevant.

Response: We reviewed the reviewers’ comments. Since the reviewers did not recommend any additional references, no modifications were needed in this section. However, to clarify our response to one of the reviewers’ comments, we have added a reference from UNAIDS to our reference list (Reference 17).

4. Check the reference list for completeness and accuracy; address or replace any retracted references and report changes in the rebuttal letter.

Response: We have carefully reviewed the reference list for completeness and accuracy. All necessary corrections have been made, and missing information has been added for several articles. We also confirm that no retracted references are included in the manuscript.

B. Questions:

1. Question 3. Have the authors made all data underlying the findings in their manuscript fully available?

Reviewer #1: No

Reviewer #2: Yes

Response: Thanks for your consideration. All data underlying the findings are fully available within the manuscript and its supporting information files (S3 Table). All studies included in this umbrella review are publicly available and cited in the reference list.

2. Question 4. Is the manuscript presented in an intelligible fashion and written in standard English?

Reviewer #1: No

Reviewer #2: Yes

Response: Thanks for your consideration. We have carefully reviewed the manuscript for language clarity, grammar, and typographical errors. All necessary corrections have been made to ensure that the text is unambiguous.

Review Comments to the Author

Reviewer 1.

Abstract:

Comment: 1. Clarify: This is a systematic review, not a general review.

Response: Thank you for the comment. We have clarified that the study is an umbrella review of systematic reviews and meta-analyses. It reads: “We conducted an umbrella review of systematic reviews with meta-analyses reporting HCV prevalence in the EMR.”

Comment: 2. Line 34: “Key populations at increased risk” should be briefly clarified

Response: Thanks for your valuable comment. We clarified the term “key populations at increased risk” in the abstract by briefly defining it, avoiding listing all specific groups to maintain conciseness. It reads: “key populations at increased risk (referring to groups disproportionately affected due to their behaviors and higher vulnerability).”

Introduction:

Comment: 3. Line 70: Clarify the meaning of “incarcerated individuals.”

Response: Thanks for your comment. The term “incarcerated individuals” has been changed to “people in prison”.

Comment: 4. Line 72: EMRO – clarify what it stands for; check if “O” is added by mistake

Response: Thanks for your comment. We have corrected the abbreviation to EMR (Eastern Mediterranean Region) to accurately refer to the geographic region rather than the WHO office. We also removed “O,” and now it reads “EMR”.

Methods:

Comment: 5. Lines 117–118: Clarify definitions of female sex workers (FSW), people living with HIV, and street children, and add supporting references.

Response: Thanks for your comment. We have clarified the definitions of these groups in the Methods section and added supporting references for them. It reads: “female sex workers (FSW; women who receive money or goods in exchange for sexual services), people living with HIV (PLHIV; individuals diagnosed with HIV), and street children (children and adolescents who live and/or work on the streets with limited or no parental care)”.

Comment: 6. Line 141: Add “of HCV” after prevalence.

Response: Thanks for your consideration. The phrase has been revised to “prevalence of HCV” for clarity.

Comment: 7. Line 145: Complete all 11 items (currently only 5 mentioned).

Response: Thanks for your comment. We have now included all 11 AMSTAR assessment items in the reference to the AMSTAR article. (Shea BJ, Grimshaw JM, Wells GA, Boers M, Andersson N, Hamel C, et al. Development of AMSTAR: a measurement tool to assess the methodological quality of systematic reviews. BMC Medical Research Methodology. 2007;7(1):10.)

It reads: “To assess the methodological quality of the included systematic reviews, two authors (ZA and PJA) independently evaluated the articles using the original AMSTAR tool (A Measurement Tool to Assess Systematic Reviews), which consists of 11 items evaluating key aspects of review methodology as described by Shea et al (19).”

Comment: 8. Line 154: Clarify the meaning of p < 0.10 (substantial heterogeneity defined as I² > 50% or p < 0.10 for the Q test).

Response: Thanks for your comment. We clarified that the p-value refers to Cochran’s Q test for heterogeneity in the revised manuscript. It reads: “Heterogeneity across studies was evaluated using the I² statistic and Cochran's Q test. In line with established recommendations for meta-analyses of observational studies, we considered heterogeneity to be substantial when I² exceeded 50%, or the Q test yielded a p-value below 0.10 (Higgins & Thompson, 2002; Huedo-Medina et al., 2006) (20). Heterogeneity and pooled prevalence are presented in S3 Tables in the supplementary information.”

Results

Comment: 9. Line 188: Clarify the meaning of (key population (20–46)).

Response: The term “key populations” has been clarified in the Results section, and reference to the definition provided in the Methods section has been added. It reads: “key populations (including PWID, people who use drugs, MSM, FSW, PLHIV, and street children) contributed 76 estimates (35.9%; references (22-48))”(1).

Comment: 10. Line 201: Correct interpretation: “CCA reached 5.4%, indicating moderate overlap” (not slight).

Response: Thanks for your consideration. We checked it again, and the interpretation of the CCA value has been corrected to indicate moderate overlap. It reads: “In key populations, the CCA reached 5.4%, indicating moderate overlap(2).”

Comment: 11. Line 226: Co-infected group not mentioned in the temporal shift in prevalence — add if relevant.

Response: Thanks for your valuable comments. We have clarified that insufficient data were available to assess temporal shifts in the co-infected group.

References

Comment: 12. Numbers 13, 46, and 58: Add full citation details (volume, page numbers, etc.).

Response: Full citation details, including volume, issue, and page numbers, have been checked and added for all references, and the missed items have been fixed.

English editing

Comment: 13. Needed throughout. Example: Line 317 –“As a result, may partially contribute to observed differences in infection burden” → needs grammatical/clarity editing.

Response: Thanks for your comment. We have reviewed the manuscript for clarity and grammatical correctness. The sentence on line 317 has been revised. Now it reads: “As a result, it may partially contribute to the observed differences in infection burden between regions.” Similar edits have been applied throughout to improve readability.

Tables

Comment: 14. Tables are not included in the review — consider including them or clarifying.

Response: Thanks for your comment. Tables are provided in the Supporting Information files and have been clearly referenced in the manuscript.

Reviewer 2.

Comment: 1. Inconsistency in time-period grouping definition

Abstract says “median year of meta-analysis publication”; methods say “median year of data collection of primary studies”. → Align and clarify which one was actually used.

Response: Thank you for this comment. The time-period grouping was based on the median year of data collection of primary studies. The abstract has been revised to ensure consistency with the Methods section. Now it reads: “For time-period analyses, meta-analyses were grouped into two intervals (before 2015 / 2015 and after) based on the median year of data collection of primary studies.”

Comment: 2. Search date clarification

Stated search up to October 12, 2025 (likely a typo for 2024 or earlier). Confirm and correct.

Response: Thank you for this observation. To ensure that no newly published studies were missed, we updated the literature search before submission. Therefore, the final search date was October 12, 2025, given that the manuscript was submitted in January 2026.

Comment: 3. Minor numerical discrepancies / rounding. e.g., key populations reported as 32% in one place, 31% in another → standardize presentation (preferably as percentages with consistent CI formatting).

Response: Thank you for catching this typo. We have reviewed and standardized all numerical values and confidence interval formatting throughout the manuscript to ensure consistent reporting.

Comment: 4. Quality assessment details

Response: Thanks for your comment. Details of the quality assessment tool (AMSTAR) are described in the Methods section, and the quality rating of each included study (high, moderate, or low) is provided in the supporting information (S3 Table). It reads: “To assess the methodological quality of the included systematic reviews, two authors (ZA and PJA) independently evaluated the articles using the original AMSTAR tool (A Measurement Tool to Assess Systematic Reviews), which consists of 11 items evaluating key aspects of review methodology as described by Shea et al (19). Each item was rated as 'Yes', 'No', or 'Not applicable', yielding a total score ranging from 0 to 11. Scores of 8–11 were classified as high quality, 4–7 as moderate quality, and 0–3 as low quality. Any discrepancies between the two reviewers were resolved through discussion. Studies with sufficient quality were selected for additional investigation. The results of quality ratings for each study are provided in the supporting information (S3 Table). Only studies with moderate or high quality (AMSTAR score ≥4) were included in further analyses, while studies with low quality (AMSTAR score 0–3) were excluded. Excluding low-quality meta-analyses helps reduce potential bias and ensures that the findings of the umbrella review are based on more reliable evidence.”

Comment: 5. Explicitly state AMSTAR threshold for inclusion (e.g., ≥4) and discuss implications of excluding low-quality meta-analyses.

Response: Thanks for your comment. We have explicitly stated the AMSTAR threshold for inclusion (score ≥4) and added some information on the implications of excluding low-quality meta-analyses in the revised manuscript. It reads: “Each item was rated as 'Yes', 'No', or 'Not applicable', yielding a total score ranging from 0 to 11. Scores of 8–11 were classified as high quality, 4–7 as moderate quality, and 0–3 as low quality. Any discrepancies between the two reviewers were resolved through discussion. Studies with sufficient quality were selected for additional investigation. The results of quality ratings for each study are provided in the supporting information (S3 Table). Only studies with moderate or high quality (AMSTAR score ≥4) were included in further analyses, while studies with low quality (AMSTAR score 0–3) were excluded. Excluding low-quality meta-analyses helps reduce potential bias and ensures that the findings of the umbrella review are based on more reliable evidence.”

Comment: 6. Heterogeneity & publication bias

Response: Thanks for your valuable comment. We have clarified the methods used to assess heterogeneity and publication bias. Heterogeneity values were extracted from the included meta-analyses. The results are presented in the supporting information (S3 Table). Publication bias was assessed qualitatively, consistent with recommendations for meta-analyses of proportions. A random-effects model was applied to account for variations across populations.

It reads: “Heterogeneity across studies was evaluated using the I² statistic and Cochran's Q test. In line with established recommendations for meta-analyses of observational studies, we considered heterogeneity to be substantial when I² exceeded 50% or the Q test yielded a p-value below 0.10(3). Heterogeneity and pooled prevalence are presented in S3 Tables in the supporting information. Publication bias testing, including funnel plot and associated tests, was not conducted, as they do not yield reliable results for meta-analysis of proportions due to the bounded nature and instability of variance in proportion data. Instead, publication bias was assessed qualitatively. This approach is consistent with methodological recommendations for meta-analyses of prevalence(4). The random-effects model accounts for expected heterogeneity across countries with different population sizes and prevents studies with very large samples from disproportionately influencing the pooled estimate. Stata software version 17.0 (Stata Corporation, College Station, TX, USA) was used to perform all statistical analyses.”

Comment: 7. Inconsistent use of “EMR” vs “EMRO”.

Response: Thanks for your comment. We have standardized the manuscript to use “EMR” consistently throughout, as it refers to the geographic region and its countries.

References:

1. UNAIDS terminology guidelines [Internet]. 2024. Available from: https://www.unaids.org/sites/default/files/media_asset/2024-terminology-guidelines_en.pdf.

2. Kirvalidze M, Abbadi A, Dahlberg L, Sacco LB, Calderón‐Larrañaga A, Morin L. Estimating pairwise overlap in umbrella reviews: considerations for using the corrected covered area (CCA) index methodology. Research Synthesis Methods. 2023;14(5):764–7.

3. Huedo-Medina TB, Sánchez-Meca J, Marin-Martinez F, Botella J. Assessing heterogeneity in meta-analysis: Q statistic or I² index? Psychological methods. 2006;11(2):193.

4. Hunter JP, Saratzis A, Sutton AJ, Boucher RH, Sayers RD, Bown MJ. In meta-analyses of proportion studies, funnel plots were found to be an inaccurate method of assessing publication bias. Journal of clinical epidemiology. 2014;67(8):897–903.

---

## [Decision Letter · Decision Letter 1]

24 Mar 2026

Hepatitis C distribution across diverse population groups in the Eastern Mediterranean Region: An Umbrella Review

PONE-D-26-00865R1

Dear Dr. Sharifi,

We’re pleased to inform you that your manuscript has been judged scientifically suitable for publication and will be formally accepted for publication once it meets all outstanding technical requirements.

Kind regards,

Ashraf Elbahrawy

Academic Editor

PLOS One

Additional Editor Comments (optional):

Reviewers' comments:

Reviewer's Responses to Questions

**Comments to the Author**

Reviewer #1: All comments have been addressed

Reviewer #2: (No Response)

2. Is the manuscript technically sound, and do the data support the conclusions?

Reviewer #1: Yes

Reviewer #2: Yes

3. Has the statistical analysis been performed appropriately and rigorously?

Reviewer #1: Yes

Reviewer #2: Yes

4. Have the authors made all data underlying the findings in their manuscript fully available?

Reviewer #1: Yes

Reviewer #2: Yes

5. Is the manuscript presented in an intelligible fashion and written in standard English?

Reviewer #1: Yes

Reviewer #2: (No Response)

Reviewer #1: (No Response)

Reviewer #2: (No Response)

.

Reviewer #1: **Yes:** Sadek MostafaSadek MostafaSadek MostafaSadek Mostafa

Reviewer #2: **Yes:** Ali MadianAli MadianAli MadianAli Madian

---

## [Editor Report · Acceptance letter]

PONE-D-26-00865R1

PLOS One

Dear Dr. Sharifi,

I'm pleased to inform you that your manuscript has been deemed suitable for publication in PLOS One. Congratulations! Your manuscript is now being handed over to our production team.

Kind regards,

on behalf of

Prof. Ashraf Elbahrawy

Academic Editor

PLOS One